# Supramolecular Assemblies in Pyridine- and Pyrazole-Based Coordination Compounds of Co(II) and Ni(II): Characterization, Hirshfeld Analysis and Theoretical Studies

**Trishnajyoti Baishya** [1], **Rosa M. Gomila** [2], **Miquel Barcelo-Oliver** [2], **Diego M. Gil** [3], **Manjit K. Bhattacharyya** [1,*] **and Antonio Frontera** [2,*]

1  Department of Chemistry, Cotton University, Assam 781001, India
2  Department de Química, Universitat de les IllesBalears, Crta de Valldemossa km 7.7, 07122 Palma de Mallorca, Spain
3  INBIOFAL (CONICET—UNT), Instituto de Química Orgánica, Facultad de Bioquímica, Química y Farmacia, Universidad Nacional de Tucumán, Ayacucho 471, San Miguel de Tucumán T4000CAN, Argentina
*  Correspondence: manjit.bhattacharyya@cottonuniversity.ac.in (M.K.B.); toni.frontera@uib.es (A.F.)

**Abstract:** Two new Ni(II) and Co(II) coordination compounds, viz., [Ni(H$_2$O)$_5$(DMAP)](IPhth)·2H$_2$O (**1**) and [Co(Hdmpz)$_4$(H$_2$O)$_2$]Cl$_2$ (**2**) (where DMAP = 4-dimethylaminopyridine, IPhth = Isophthalate, Hdmpz = 3,5-dimethylpyrazole),were synthesized and characterized using elemental analysis, TGA, spectroscopic (FTIR and electronic) and single-crystal X-ray diffraction techniques. Compound **1** crystallizes as a co-crystal hydrate of Ni(II), whereas compound **2** is a mononuclear compound of Co(II). The crystal structure analysis of compound **1** reveals the presence of various non-covalent interactions such as anion–π, π–π, C–H···π, C–H···C, etc., which stabilize the layered assembly of the compound. In compound **2**, enclathration of counter chloride ions within the supramolecular trimeric host cavity plays a crucial role in the stabilization of the compound. The non-covalent interactions observed in the crystal structures were further studied theoretically, focusing on the cooperative π-stacking interactions between the *DMAP* and *IPhth* counter-ions in **1**. To identify the non-covalent interactions of the compounds, Hirshfeld surfaces and their associated two-dimensional fingerprint regions were analyzed. Theoretical calculations confirm that H-bonding interactions combined with the π-stacking contacts are crucial synthons for the solid-state stability of compound **1**.

**Keywords:** co-crystal hydrate; supramolecular assemblies; enclathration; cooperative π-stacking; DFT calculations

## 1. Introduction

Transition-metal-based coordination compounds involving organic ligands have attracted researchers because of their wide potential applications in catalysis, non-linear optics, semiconductor devices and in biology [1–4]. The intriguing self-assemblies of metal–organic compounds have also gained remarkable interest because of their interesting structural topologies and network architectures [5,6]. Metal–organic co-crystals, involving metal centers having a minimum of two components with lattice water molecules, have also received emphasis owing to their wide practical utilities in the pharmaceutical industry, electronic devices as well as in synthetic chemistry [7,8].

Over the last few years, various N-donors, especially with pyridine-based moieties, have been extensively studied to construct supramolecular assemblies of desired architectures and potential applications [9,10]. Similarly, organic moieties containing pyrazole rings have also been exploited to explore self-assembled architectures with fascinating properties [11,12]. Metal–organic compounds of pyrazole- and pyridine-based ligands have been reported to display interesting biological applications [13,14]. The multiple coordination modes of the carboxylate-based aromatic ligands such as terminal, bridging,

chelating, syn-anti, anti-anti, etc., have resulted in the formation of metal–organic compounds with fascinating structural topologies [15–20]. Non-covalent interactions are the backbone of supramolecular chemistry which play interesting roles in various fields such as synthesis, catalysis, crystal engineering, design of pharmaceutical agents, molecular biology, etc. [21,22]. Non-covalent contacts involving aromatic rings play crucial roles in the solid-state stability of metal–organic compounds [23] and also in guiding structural topologies of novel functional materials [24,25]. Counter anions present in the crystal lattice of metal–organic compounds not only neutralize the positive charge of complex cationic moieties but are also involved in non-covalent interactions, adding new dimensions in crystal engineering [26,27].

In the present study; we aim to explore the synthesis, detailed structural investigations and supramolecular assemblies of two new Ni(II) and Co(II) compounds involving pyridine and pyrazole ligands. The compounds were characterized using FT-IR, electronic spectroscopic techniques, TGA and elemental analysis. The presence of various non-covalent contactsin the crystal structure of the compounds, such as anion–$\pi$, $\pi$–$\pi$, C–H$\cdots\pi$, C–H$\cdots$C, etc., stabilize the layered assemblies. In compound **2**, enclathration of counter chloride ions within the supramolecular trimeric host cavity plays an important role in the stabilization of the compound. The non-covalent interactions in the crystal structures have been studied using Hirshfeld surface analysis and density functional theory (DFT) calculations, focusing on the $\pi$-stacking interactions between the *DMAP* and *IPhth* counter-ions that are further assisted by strong H-bonds involving the water molecules. The supramolecular ternary assembly was further characterized using molecular electrostatic potential (MEP) surface analysis and combined quantum theory of atoms-in-molecules (QTAIM) and non-covalent interaction (NCI) plot computational tools.

## 2. Materials and Methods

The chemicals used in the present study, viz., nickel(II) chloride hexahydrate, cobalt(II) chloride hexahydrate, isophthalic acid, 4-dimethylaminopyridine and 3,5-dimethyl pyrazole, were purchased from commercial sources and used without further purifications. The elemental analyses of the compounds were carried out using Perkin Elmer 2400 series II CHNS/O analyzer. The FT-IR spectra of the compounds were recorded in the frequency region 4000–500 cm$^{-1}$ using Bruker Alpha (II) Infrared spectrophotometer. The electronic spectra of the compounds were recorded using Shimadzu UV-2600 spectrophotometer. For UV-Vis-NIR spectra, BaSO$_4$ powder was used as reference (100% reflectance). Sherwood Mark 1 magnetic susceptibility balance was used to calculate the room-temperature magnetic moments of the compounds. Thermogravimetric studies were carried out under the flow of N$_2$ gas using Mettler Toledo TGA/DSC1 STAR$^e$ system at the heating rate of 10 °C min$^{-1}$.

*2.1. Syntheses*

2.1.1. Synthesis of [Ni(H$_2$O)$_5$(DMAP)](IPhth)·2H$_2$O (1)

In 10 mL of de-ionized water, disodium salt of isophthalic acid (0.210 g, 1 mmol) was dissolved in a round-bottom flask. NiCl$_2$·6H$_2$O (0.237 g, 1 mmol) was then slowly added and mechanically stirred for an hour at room temperature. After an hour, *DMAP* (0.122 g, 1 mmol) was slowly added to the reaction mixture and the mixture was kept stirring for another two hours (Scheme 1). Then, the reaction mixture was kept in cooling conditions in a refrigerator (below 4 °C), which yielded pale blue block-shaped single crystals after a few days. Yield: 0.415 g (88.11%). Anal. calcd. for C$_{15}$H$_{28}$N$_2$NiO$_{11}$: C, 38.24%; H, 5.99%; N, 5.95%. Found: C, 38.13%; H, 5.92%; N, 5.87%. IR (KBr pellet, cm$^{-1}$): 3454(br), 2817(w), 2729(w), 2124(br), 1620(s), 1586(sh), 1483(w), 1431(w), 1387(s), 1342(sh), 1224(w), 795(w), 730(w), 685(w) (s, strong; m, medium; w, weak; br, broad; sh, shoulder).

**Scheme 1.** Syntheses of the compounds **1** and **2**.

### 2.1.2. Synthesis of [Co(Hdmpz)$_4$(H$_2$O)$_2$]Cl$_2$ (2)

CoCl$_2$·6H$_2$O (0.237 g, 1mmol) and *Hdmpz* (0.304 g, 4 mmol) were dissolved in de-ionized water (10 mL) in a round-bottom flask and the solution was allowed to stir mechanically for about two hours (Scheme 1). Then, the resulting solution was kept unperturbed in cooling conditions (24 °C), from which red block-shaped crystals were obtained after several days. Yield: 0.480 g (87.27%). Anal. calcd. for C$_{20}$H$_{36}$Cl$_2$CoN$_8$O$_2$: C, 43.64%; H, 6.59%; N, 20.36%. Found: C, 43.55%; H, 6.51%; N, 20.27%. IR (KBr pellet, cm$^{-1}$): 3457 (br), 3139 (sh), 3038 (w), 2980 (w), 2920 (m), 2869 (w), 2780 (w), 2714 (w), 2596 (m), 2463 (w), 2294 (m), 2139 (w), 1615 (s), 1571 (s), 1475 (s), 1418 (s), 1285 (s), 1247 (w), 1160 (s), 1040 (s), 988 (w), 782 (s), 701 (m), 612 (m), 517 (m) (s, strong; m, medium; w, weak; br, broad; sh, shoulder).

### 2.2. Crystallographic Data Collection and Refinement

Single crystals of the compounds **1** and **2** were selected, covered with Parabar 10320 (formally known as Paratone N) and mounted on a cryoloop on a BRUKER D8 Venture diffractometer, with a Photon III 14 detector, using an Incoatec high brilliance ImS DIA-MOND Cu tube (λ = 1.54178 Å) equipped with an Incoatec Helios MX multilayer optics. The crystals were kept at 151 K (compound **1**) or 100 K (compound **2**) during data collection. Data reduction and cell refinements were performed using the Bruker APEX4 program [28]. Scaling and absorption corrections were carried out using SADABS [29]. Crystal structures were solved by direct method and refined by full-matrix least-squares technique with SHELXL-2018/3 [30] using WinGX [31] platform. All the non-hydrogen atoms were refined anisotropically. The hydrogen atoms were placed at their calculated positions and refined in the isotropic approximation, except for those in compound 2 attached to water O-atoms, which were located using a Fourier difference map and refined isotropically.

Diamond 3.2 software was used to draw all the structural diagrams [32]. Table 1 contains the crystallographic data and the structure refinement table for the compounds.

**Table 1.** Crystallographic data and structure refinement details for the compounds **1** and **2**.

| Parameters | 1 | 2 |
|---|---|---|
| Formula | C$_{15}$H$_{28}$N$_2$NiO$_{11}$ | C$_{20}$H$_{36}$Cl$_2$CoN$_8$O$_2$ |
| Formula weight | 471.10 | 550.40 |
| Temp, (K) | 151 | 100.0 |
| Crystal system | Triclinic | Monoclinic |
| Space group | $P\bar{1}$ | $C2/c$ |
| a, (Å) | 7.073(4) | 10.4681(7) |
| b, (Å) | 11.591(6) | 14.1306(10) |
| c, (Å) | 12.804(7) | 18.4697(13) |
| α, (°) | 83.465(19) | 90 |
| β, (°) | 84.37(2) | 92.765(3) |
| γ, (°) | 78.857(18) | 90 |
| V, (Å$^3$) | 1020.0(9) | 2728.9(3) |
| Z | 2 | 4 |
| Absorption coefficient (mm$^{-1}$) | 1.925 | 6.991 |
| F(0 0 0) | 496.0 | 1156.0 |
| $\rho_{calc}$g/cm$^3$ | 1.524 | 1.340 |

**Table 1.** *Cont.*

| Parameters | 1 | 2 |
|---|---|---|
| index ranges | $-8 \leq h \leq 8$ <br> $-13 \leq k \leq 14$, <br> $-14 \leq l \leq 15$ | $-12 \leq h \leq 12$, <br> $-16 \leq k \leq 16$, <br> $-22 \leq l \leq 22$ |
| Crystal size, (mm$^3$) | $0.48 \times 0.28 \times 0.25$ | $0.21 \times 0.18 \times 0.07$ |
| 2θ range, (°) | 9.956 to 138.182 | 10.524 to 136.818 |
| Independent reflections | 3697 [$R_{int} = 0.0669$, $R_{\sigma} = 0.1028$] | 2504 [$R_{int} = 0.0518$, $R_{\sigma} = 0.0227$] |
| Reflections collected | 16,188 | 36,237 |
| Refinement method | Full-matrix least-squares on F$^2$ | Full-matrix least-squares on F$^2$ |
| Data/restraints/parameters | 3697/1/278 | 2504/0/162 |
| Goodness-of-fit on F$^2$ | 1.093 | 1.147 |
| Final R indices (I > 2σ(I)) (all data) | $R_1 = 0.0414$, $wR_2 = 0.1041$ <br> $R_1 = 0.0803$, $wR_2 = 0.1072$ | $R_1 = 0.0759$, $wR_2 = 0.1804$ $R_1 = 0.0772$, <br> $wR_2 = 0.1811$ |
| Largest hole and peak (e·Å$^{-3}$) | $0.46/-0.41$ | $1.34/-1.46$ |

*2.3. Computational Methods*

2.3.1. Theoretical Study

The energies and wavefunction calculations were performed at the RI-BP86-D3/def2-TZVP [33,34] level of theory and with the Turbomole 7.2 [35] program. Only the position of the H-atoms was optimized, keeping the non-H-atoms frozen. For the octahedral Ni(II) complex, the high spin configuration was considered (triplet state). The quantum theory of atoms-in-molecules (QTAIM) and non-covalent interaction (NCI) plot [36] computational tools were used to characterize the NCIs. The QTAIM analysis [37] was performed using the MULTIWFN program [38] and represented using VMD software [39].

2.3.2. Hirshfeld Analysis

Hirshfeld surface analysis was used for studying intermolecular interactions [40–42] by means of the CrystalExplorer 21 software [43]. The normalized contact distance ($d_{norm}$) was measured using t$d_e$ (the distance from the point to the nearest nucleus external to the surface), $d_i$ (the distance to the nearest nucleus internal to the surface) and the van der Waals (vdW) radii of the atoms involved in the intermolecular contacts. The $d_{norm}$ values allow identifying the different regions participating in the intermolecular interactions. Graphical plots of the Hirshfeld surfaces mapped with $d_{norm}$ function show a color code of red (shorter contacts), white (contacts around the sum of vdW radii of atoms) and blue (longer contacts). The Hirshfeld surfaces were mapped over a $d_{norm}$ range of $-0.075$ a.u. (red) to $+0.75$ a.u. (blue), *shape index* of $-1.0$ a.u. (concave) to $+1.0$ a.u. (convex) and curvedness of $-4.00$ a.u. (flat) to $+0.4$ a.u. (singular). The 2D fingerprints were plotted using the translated 0.6–2.8 Å range, including reciprocal contacts.

**3. Results and Discussion**

*3.1. Syntheses and General Aspects*

[Ni(H$_2$O)$_5$(DMAP)](IPhth)·2H$_2$O (**1**) was prepared by the reaction between one equivalent of NiCl$_2$·6H$_2$O, one equivalent of *DMAP* and one equivalent of disodium salt of isophthalic acid at room temperature in de-ionized water medium. Similarly, [Co(Hdmpz)$_4$(H$_2$O)$_2$]Cl$_2$ (**2**) was synthesized by reacting one equivalent of CoCl$_2$·6H$_2$O with four equivalents of *Hdmpz* under similar reaction conditions. Both compounds **1** and **2** are soluble in water as well as in common organic solvents. Compounds **1** and **2** exhibit room-temperature (298 K) μ$_{eff}$ values of 2.86 and 3.85 BM, respectively, suggesting the presence of two and three unpaired electrons in the Ni(II) and Co(II) centers of the distorted octahedral coordination spheres of **1** and **2**, respectively [44,45].

### 3.2. Crystal Structure Analysis

Figure 1 depicts the molecular structure of compound **1**. Table 2 contains the details of the selected bond lengths and bond angles around the Ni(II) center of compound **1**. Compound **1** crystallizes in the triclinic *P*$\overline{1}$ space group. The Ni(II) center of compound **1** is coordinated to five coordinated aqua ligands and one *DMAP* moiety. The dipositive charge of the complex cationic moiety of compound 1 is neutralized by the presence of one uncoordinated *Iphth* moiety in the crystal lattice. In addition, the asymmetric unit of the compound comprises two uncoordinated water molecules. The coordination geometry around the Ni(II) center is distorted octahedron, where the axial sites are occupied by one coordinated water molecule (O1W) and N1 from the *DMAP* moiety; whereas the equatorial sites are occupied by the remaining four coordinated water molecules (O5W, O2W, O3W and O4W). The equatorial atoms, viz., O5W, O2W, O3W and O4W, are distorted from the equatorial plane with the mean r.m.s. deviation of 0.0304 Å. The average Ni–O and Ni–N bond lengths are well consistent with similar Ni(II) compounds [46,47].

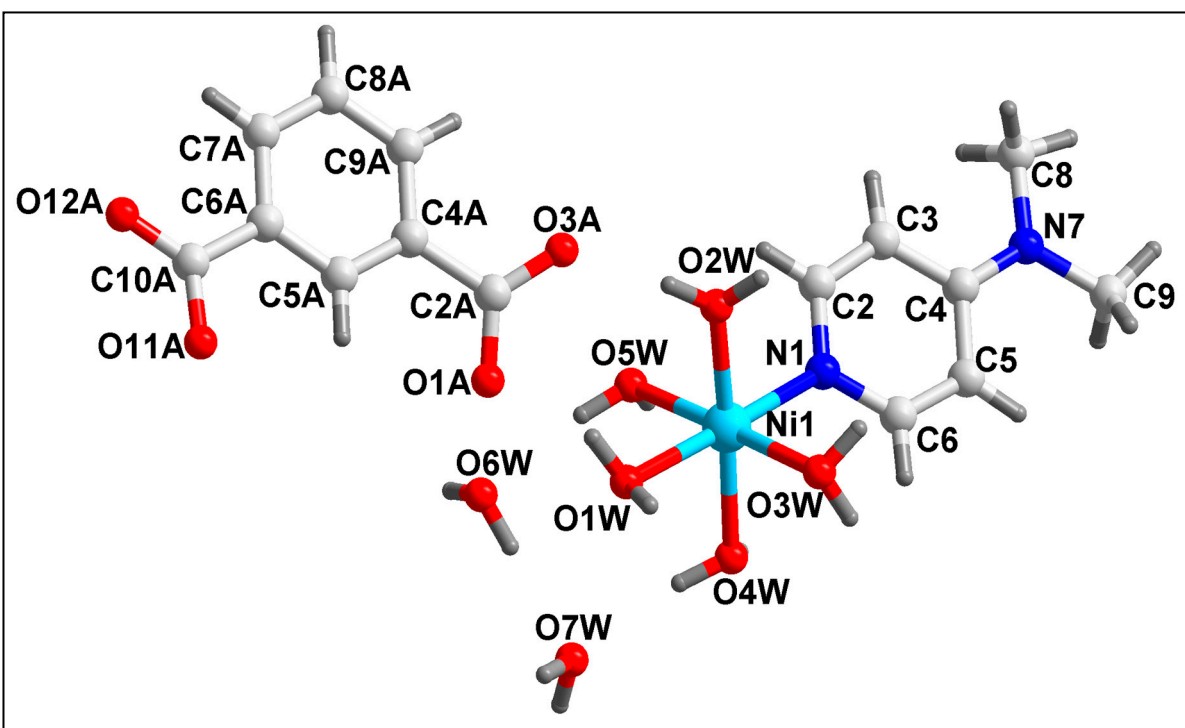

**Figure 1.** Molecular structure of [Ni(H$_2$O)$_5$(DMAP)](IPhth)·2H$_2$O (**1**). Aromatic hydrogen atoms are omitted for clarity.

Figure 2 represents the supramolecular trimer formed in the crystal structure of compound **1**, which is stabilized by O–H···O, C–H···O hydrogen bonding, non-covalent C–H···π, anion-π and π-π interactions. As shown in Figure 2, the O–H···O hydrogen bonding interactions are observed between the coordinated water molecules (O2W and O3W) and carboxyl O atoms (O11A and O12A) of uncoordinated *IPhth* moieties having O2W–H2WB···O12A and O3W–H3WA···O11A distances of 1.91 and 1.90 Å, respectively (Table 3). The carboxyl O atom (O1A) is involved in C–H···O hydrogen bonding interactions with the –CH moiety of coordinated *DMAP* having a C9–H9C···O1Adistance of 2.80 Å. The C–H···π interaction [47] is observed between the –CH moiety of coordinated *DMAP* and the aromatic ring of the uncoordinated *IPhth* moiety having C9···Cg and H9A···Cg distances of 3.913(3) and 3.566 Å, respectively (Cg is the ring centroid defined by the atoms C4A-C8A). The corresponding C9–H9A···Cg angle is found to be 108°. The existence of this interaction was further confirmed by combined QTAIM/NCI plot computational tools (vide infra).

**Table 2.** Selected bond lengths (Å) and bond angles (°) of Ni(II) and Co(II) centers in **1** and **2**, respectively.

| Compound 1 | | | |
|---|---|---|---|
| Ni1–O1W | 2.0912(2) | O2W–Ni1–O1W | 89.52(7) |
| Ni1–O2W | 2.0297(2) | O2W–Ni1–O3W | 89.15(7) |
| Ni1–O3W | 2.1004(2) | O2W–Ni1–O4W | 174.49(7) |
| Ni1–O4W | 2.0778(2) | O2W–Ni1–O5W | 91.25(7) |
| Ni1–O5W | 2.1043(2) | O2W–Ni1–N1 | 90.137) |
| Ni1–N1 | 2.059(2) | O3W–Ni1–O1W | 88.67(7) |
| O4W–Ni1–O1W | 85.31(6) | O4W–Ni1–O3W | 88.85(7) |
| O4W–Ni1–O5W | 90.59(7) | O5W–Ni1–O1W | 89.58(7) |
| O5W–Ni1–O3W | 178.20(6) | N1–Ni1–O1W | 178.25(6) |
| N1–Ni1–O3W | 93.03(7) | N1–Ni1–O4W | 95.10(7) |
| N1–Ni1–O5W | 88.72(7) | | |
| Compound 2 | | | |
| Co1–O1W | 2.076(5) | O1W–Co1–N1 | 91.03(1) |
| Co1–O2W | 2.062(6) | O1W–Co1–N1#1 | 91.03(1) |
| Co1–N1#1 | 2.110(5) | O1W–Co1–N6 | 89.57(1) |
| Co1–N1 | 2.110(5) | O1W–Co1–N6#1 | 89.57(1) |
| Co1–N6 | 2.105(5) | O2W–Co1–O1W | 180.0 |
| Co1–N6#1 | 2.105(5) | O2W–Co1–N1#1 | 88.97(1) |
| O2W–Co1–N1 | 88.97(1) | O2W–Co1–N6#1 | 90.43(1) |
| O2W–Co1–N6 | 90.43(1) | N1#1–Co1–N1 | 177.9(3) |
| N6–Co1–N1#1 | 88.32(2) | N6#1–Co1–N1 | 88.32(2) |
| N6#1–Co1–N1#1 | 91.70(2) | N6–Co1–N1 | 91.70(2) |
| N6–Co1–N6#1 | 179.1(3) | | |

#1 1 − X, +Y, 3/2 − Z.

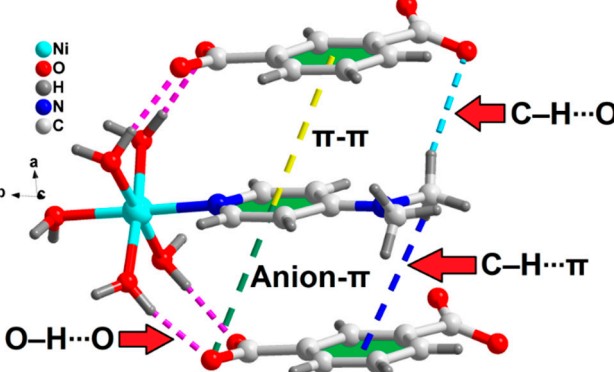

**Figure 2.** Formation of a supramolecular trimer in compound **1** assisted by O–H⋯O, C–H⋯O hydrogen bonding, non-covalent C–H⋯π, anion-π and π-π interactions.

Moreover, a closer look reveals the presence of a strong anion-π interaction, which is observed between the aromatic ring of the coordinated *DMAP* moiety and the carboxyl O (O12A) atom of the uncoordinated *IPhth* moiety. The distance between the centroid of the aromatic ring and the O12A atom is found to be 3.99(2)Å. The angle between O12A, the ring centroid (Cg) and the C2 atom (defining the aromatic plane) is found to be 95.64°, which suggests the strong nature of the anion-π interaction [48]. Moreover, aromatic π-stacking interactions are also observed between the phenyl and pyridyl rings of uncoordinated *IPhth* and coordinated *DMAP* moieties, respectively, having the centroid(C4A, C5A, C6A, C7A, C8A, C9A)–centroid(C2, C3, C4, C5, C6, N1) distance of 3.857(1) Å. The corresponding slipped angle, the angle between the ring normal and the vector between two ring centroids, is found to be 20.6(2)° [49]. These non-covalent interactions have been further studied theoretically (vide infra). These supramolecular trimers propagate along the crystallographic *a* axis to form the 1D supramolecular chain of the compound (Figure 3).

**Table 3.** Selected hydrogen bond distances (Å) and angles (deg.) for compounds **1** and **2**.

| D–H···A | $d$(D–H) | $d$(D···A) | $d$(H···A) | <(DHA) |
|---|---|---|---|---|
| Compound **1** | | | | |
| O2W–H2WB···O12A#1 | 0.87 | 2.741(2) | 1.91 | 160.3 |
| O3W–H3WA···O11A#1 | 0.87 | 2.767(2) | 1.90 | 172.7 |
| O5W–H5WB···O12A#2 | 0.87 | 2.821(2) | 1.95 | 177.1 |
| O4W–H4WA···O11A#2 | 0.87 | 2.762(2) | 1.91 | 165.9 |
| O5W–H5WA···O6W | 0.87 | 2.776(3) | 1.92 | 168.4 |
| O7W–H7WA···O6W#3 | 0.87 | 2.824(3) | 1.96 | 174.4 |
| O4W–H4WB···O7W | 0.87 | 2.837(2) | 1.98 | 168.8 |
| Compound **2** | | | | |
| O1W–H1W···Cl1#4 | 1.03(7) | 3.039(3) | 2.14(7) | 167(6) |
| O2W–H2W···Cl1 | 0.91(7) | 3.030(3) | 2.14(7) | 169(7) |

#1 + X, −1 + Y, +Z, #2 1 + X, −1 + Y, +Z, #3 1 − X, 1 − Y, 1 − Z #4 1/2 − X, −1/2 + Y, 3/2 − Z.

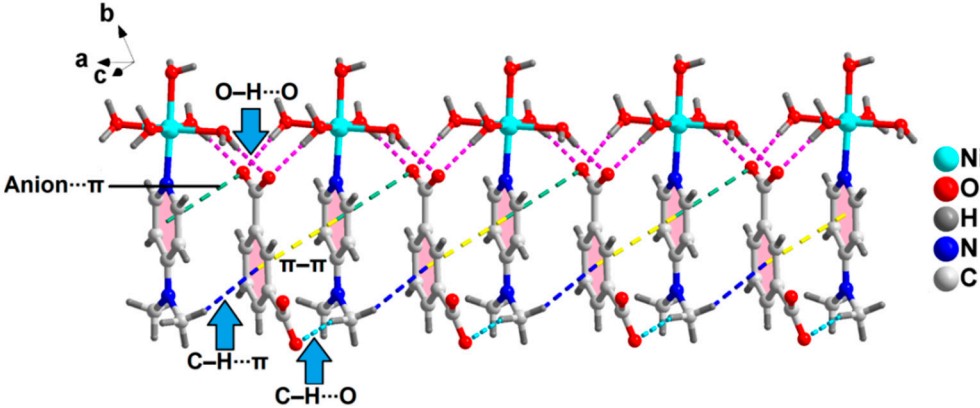

**Figure 3.** The1D supramolecular chain of compound **1** along the crystallographic *a* axis.

The adjacent 1D supramolecular chains of compound **1** are interconnected via O–H···O and C–H···O hydrogen bonding interactions to form the layered architecture along the crystallographic *ab* plane (Figure S1). The lattice water molecule O6W is involved in O–H···O hydrogen bonding interactions with a coordinated water molecule (O5W) and a carboxyl O atom (O1A) having O5W–H5WA···O6W and O6W–H6WB···O1A distances of 1.92 and 1.96 Å, respectively. Carboxyl O atoms (O1A and O3A) are also involved in O–H···O hydrogen bonding interactions with coordinated water molecules (O2W and O1W) having O2W–H2WA···O1A and O1W–H1WA···O3A distances of 1.80 and 1.95 Å, respectively.

A detailed structural investigation reveals the formation of the supramolecular layered assembly of **1** along the crystallographic *bc* plane assisted by the lattice water molecules. As shown in Figure 4, water molecules (O6W and O7W) are involved in O–H···O hydrogen bonding contacts with two coordinated aqua molecules (O5W and O4W) from two neighboring dimers.

The O5W–H5WA···O6W and O4W–H4WB···O7W distances are found to be 1.91 and 1.97 Å, respectively. Moreover, the –CH moiety of the pyridyl ring of *DMAP* is involved in C–H···O hydrogen bonding interactions with the coordinated aqua molecule (O4W) having a C6–H6···O4W distance of 2.54 Å. Figure 5 depicts the molecular structure of compound 2. Table 2 contains the details of the selected bond lengths and bond angles around the Co(II) center. Compound **2** crystallizes in the monoclinic *C2/c* space group. The Co(II) center of compound **2** is hexa-coordinated with four coordinated *Hdmpz* moieties and two coordinated water molecules. The presence of two uncoordinated chloride ions compensates for the dipositive charge of the cationic complex moiety. The X-ray crystallographic analysis reveals that the mononuclear Co(II) complex moiety possesses a two-fold axis of symmetry

which passes through the coordinated water molecules (O1W and O2W) and the Co(II) center of the compound. The coordination geometry around the Co(II) center is distorted octahedron, where the axial sites are occupied by the two coordinated water molecules (O1W and O2W) and the equatorial sites are occupied by N1, N1′, N6 and N6′ atoms of the four coordinated *Hdmpz* moieties. The four equatorial atoms, viz., N1, N1′, N6 and N6′, are distorted from the mean equatorial plane with the mean r.m.s. deviation of 0.0046 Å. The average Co–O and Co–N bond lengths are almost consistent with the previously reported Co(II) complexes [50,51].

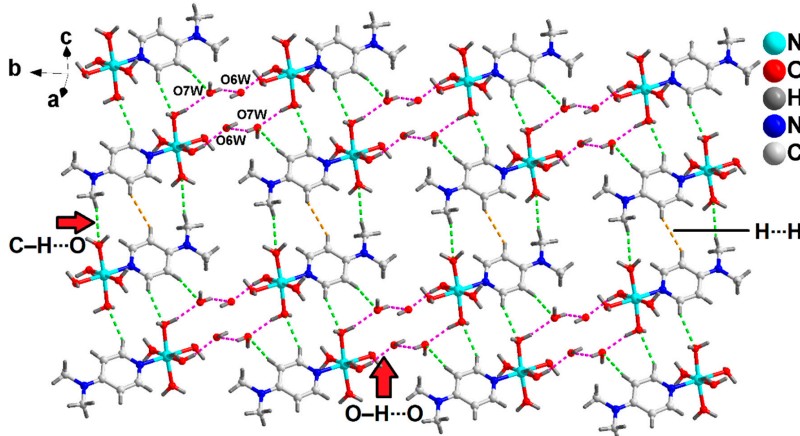

**Figure 4.** Layered assembly of compound **1** along the crystallographic *bc* plane assisted by lattice water molecules.

As shown in Figure S2, the adjacent monomeric units of compound **2** are involved in the formation of a supramolecular 1D chain stabilized by non-covalent C–H⋯C interactions between the neighboring monomeric unit having C9–H9⋯C4 and C4–H4⋯C9 bond distances of 2.95 and 3.56 Å, respectively, [C(sp$^2$)–H⋯C(sp$^2$), C9⋯C4 = 3.75(2) Å]. In addition, C–H⋯C interactions are also observed between the –CH moieties of methyl groups of *Hdmpz* of one monomeric unit and a C atom of *Hdmpz* of a neighboring monomeric unit having C101–H10B⋯C51 and C51–H51B⋯C101 distances of 3.11 and 3.19 Å, respectively, [C(sp$^3$)–H⋯C(sp$^3$), C101⋯C51 = 3.89(1) Å].

The adjacent 1D supramolecular chains of compound **2** are interconnected via non-covalent C–H⋯C and C–H⋯π interactions to form the layered architecture along the crystallographic *ac* plane (Figure 6). The –CH moieties of methyl groups of *Hdmpz* are involved in C–H⋯C interactions with one C atom of methyl groups having C81–H81C⋯C31 and C31–H31C⋯C81 distances of 3.68 and 3.76 Å, respectively, [C(sp$^3$)–H⋯C(sp$^3$), C81⋯C31 = 3.94 Å]. Moreover, C-H⋯π interactions are also observed involving the H9 atom of the pyrazole moiety and the N1/N2/C3-C5 centroid having H9⋯centroid separation of 3.417 Å. The corresponding C–H⋯π angle is found to be 151.9(3)°.

The interesting aspect of the crystal structure of compound **2** is the formation of a trimeric supramolecular host cavity assisted by a number of non-covalent C–H⋯C interactions. The methyl groups of the *Hdmpz* are involved in various non-covalent C–H⋯C interactions: [C101–H10A⋯C81 = 3.29 Å, C(sp$^3$)–H⋯C(sp$^3$), C101⋯C81 = 3.68 Å; C51–H51B⋯C101 = 3.19 Å, C(sp$^3$)–H⋯C(sp$^3$), C51⋯C101 = 3.89 Å; C101–H10B⋯C51 = 3.11 Å, C(sp$^3$)–H⋯C(sp$^3$), C101⋯C51 = 3.89 Å; C51–H51B⋯C101 = 3.19 Å, C(sp$^3$)–HB⋯C(sp$^3$), C51⋯C101 = 3.89 Å; C51–H51B⋯C51 = 3.69 Å, C(sp$^3$)–H⋯C(sp$^3$), C51⋯C51 = 3.92 Å]. In addition, the –CH moieties of pyrazole ring of *Hdmpz* are also involved in C–H⋯C interactions with the C4 atom of the pyrazole ring of *Hdmpz* of the neighboring monomeric unit having a C9–H9⋯C4 distance of 2.95 Å, [C(sp$^2$)–H⋯C(sp$^2$), C9⋯C4 = 2.95 Å]. However, the most fascinating aspect of the crystal structure of compound **2** is the enclathration of the counter chloride ion (Cl1) within the supramolecular trimeric host cavity. Cl1 is

interconnected to three monomeric units via C–H···Cl, N–H···Cl and O–H···Cl hydrogen bonding interactions (Figure 7).

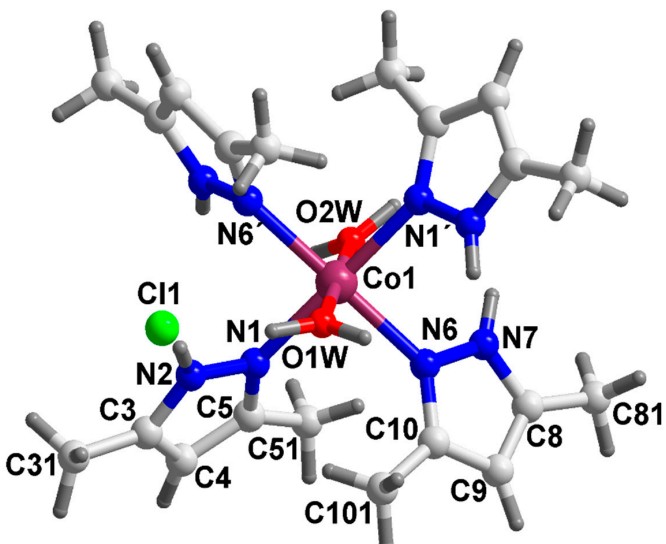

**Figure 5.** Molecular structure of [Co(Hdmpz)$_4$(H$_2$O)$_2$]Cl$_2$ (**2**). Symmetry codes for N1′ and N6′: 1 − X, +Y, 3/2 − Z.

The counter chloride ion (Cl1) is involved in C–H···Cl hydrogen bonding interactions with the –CH moieties of methyl groups of coordinated *Hdmpz*(C51–H51A···Cl1 = 2.81 Å; C31–H31A···Cl1 = 2.99 Å; C101–H10A···Cl1 = 2.74 Å). N–H···Cl hydrogen bonding interactions are found between the –NH moieties of the pyrazole ring of the coordinated *Hdmpz* and Cl1 atom of the lattice (N7–H7···Cl1 = 2.37 Å, N2–H2···Cl1 = 2.38 Å). Moreover, O–H···Cl interactions between the –OH moieties of coordinated water molecules and the lattice Cl1 (O1W–H1W···Cl1 = 2.05 Å; O2W–H2W···Cl1 = 2.08 Å) are also found to be involved in the enclathration of the counter chloride ion (Cl1) within the supramolecular trimeric host cavity. These enclathrated chloride ions within the supramolecular host cavities stabilize the layered assembly of the compound along the crystallographic *bc* plane (Figure 8).

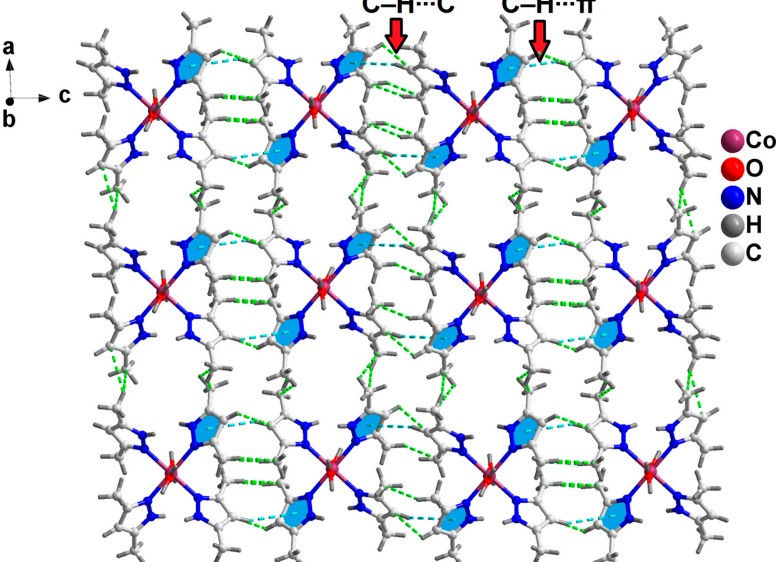

**Figure 6.** Layered architecture of compound **2** along the crystallographic *ac* plane assisted by non-covalent C–H···C, C–H···π interactions.

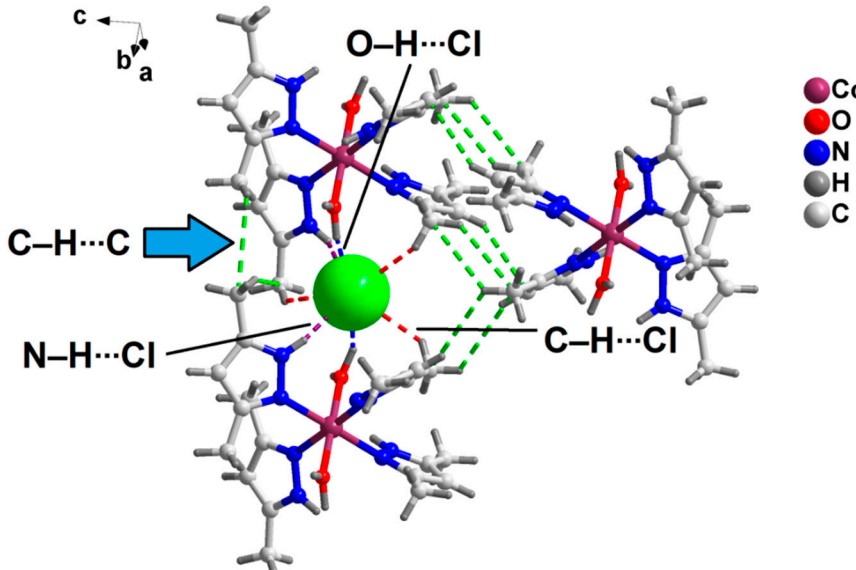

**Figure 7.** Enclathration of single guest chlorine atom within the supramolecular trimeric host cavity of compound **2**.

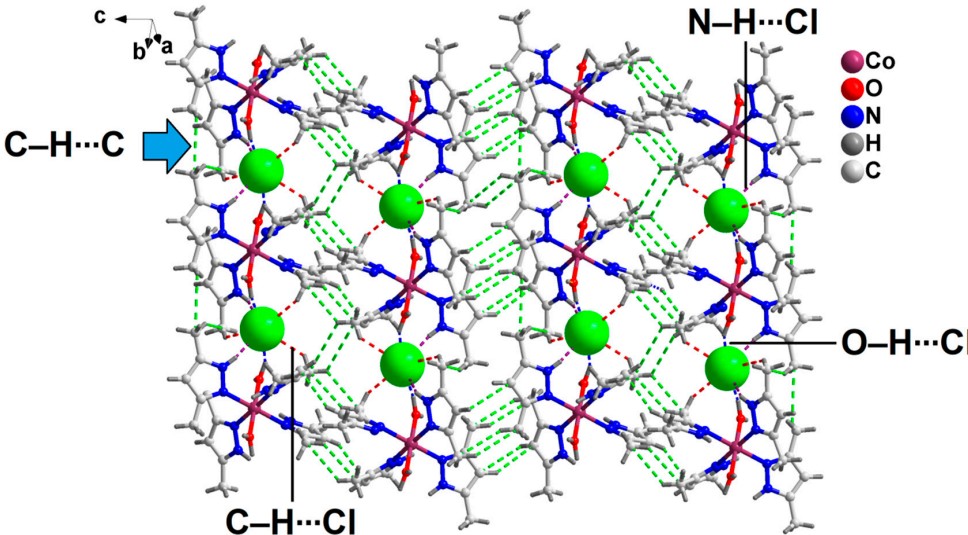

**Figure 8.** Layered assembly of compound **2** along the crystallographic *bc* plane assisted by the enclathrated chlorine atoms.

### 3.3. Spectral Studies

### 3.3.1. FT-IR Spectroscopy

The FT-IR spectra of compounds **1** and **2** have been recorded in the region $4000–500 \text{ cm}^{-1}$ (Figure S3). The broad absorption bands at around $3454 \text{ cm}^{-1}$ in the FT-IR spectra of the compounds can be correlated to the O-H stretching vibrations of the coordinated and/or lattice water molecules [52]. The FT-IR spectra of **1** and **2** also exhibit bands corresponding to $\rho_r$ ($H_2O$) ($712 \text{ cm}^{-1}$) and $\rho_w$ ($H_2O$) ($635 \text{ cm}^{-1}$), which indicate the presence of coordinated water molecules [53]. The bands at around 1620 and $1342 \text{ cm}^{-1}$ in **1** can be assigned to the asymmetric $\nu_{as}$(COO) and symmetric $\nu_s$(COO) stretching vibrations of the carboxylate moiety of the lattice *IPhth* molecule [54]. The presence of the coordinated *DMAP* moiety in compound **1** produces a weak-intensity absorption band in the region $1230–1218 \text{ cm}^{-1}$ [55]. For compound 1, the ring stretching vibrations of DMAP have been shifted to 1473, $1593 \text{ cm}^{-1}$, thereby supporting the coordination of *DMAP* through the nitrogen atom of the pyridine ring [56]. Pyridine ring vibrations of coordinated *DMAP*

are observed at 1586 and 1387 cm$^{-1}$ [57], while the ring wagging vibrations of *DMAP* are also observed at 685 and 730 cm$^{-1}$ [58]. In the FT-IR spectrum of compound **2**, the band at around 3139 cm$^{-1}$ can be assigned to the $\nu$(N–H) vibrations of the coordinated *Hdmpz* moieties [59], while the $\nu$(C–H) vibrations of the methyl groups of coordinated *DMAP* and *Hdmpz* in compounds **1** and **2** are observed in the region of 2970–2770 cm$^{-1}$ [60]. The absorption bands for C–N, N–N and C=N stretching vibrations of Hdmpz rings are obtained at 1418, 1285 and 1160 cm$^{-1}$ in **2** [61].

### 3.3.2. Electronic Spectroscopy

The electronic spectra of compounds **1** and **2** in solid as well as in aqueous phases are discussed in detail (see ESI, Figures S4 and S5). The electronic spectra of compounds **1** and **2** corroborate the presence of distorted octahedral Ni(II) and Co(II) centers in the compounds, respectively [62,63]. The peaks for $\pi \rightarrow \pi^*$ transitions of the aromatic ligands are obtained at the expected positions [64].

### 3.4. Thermogravimetric Analysis

The thermogravimetric curves of compounds **1** and **2** were obtained in the temperature range 25–800 °C under the N$_2$ atmosphere at the heating rate of 10 °C/min (Figure S6). For compound **1**, the thermal decomposition of 3.85% (calcd. = 3.82%) in the temperature range 30–80 °C can be attributed to the loss of one uncoordinated water molecule [65]. In the temperature range 81–150 °C, the weight loss of 4.14% (calcd. = 3.82%) is due to the decomposition of the other uncoordinated water molecule. The coordinated water molecules decomposed in the temperature range 151–220 °C with the observed weight loss of 17.35% (calcd. = 19.1%) [65]. In the temperature range 221–290 °C, the weight loss of 23.98% (calcd. = 25.93%) is due to the decomposition of the *DMAP* moiety [66]. The *IPhth* moiety is decomposed in the temperature range 291–638 °C with the observed weight loss of 33.79% (calcd. = 35.26%) [67]. For compound **2**, the weight loss of 6.07% (calcd. = 6.54%) in the temperature range 50–210 °C can be assigned to the loss of coordinated water molecules [65]. The observed weight loss of 12.09% (calcd. = 12.88%) in the temperature range 211–250 °C can be assigned to the loss of Cl moieties [68]. Refat et al. have reported similar behavior of Cl moieties in Schiff base compounds of Cu(II), Co(II) and Ni(II) [69]. Three *Hdmpz* moieties undergo thermal decomposition in the temperature range 251–750 °C with the observed weight loss of 54.12% (calcd. = 52.32%) [70].

### 3.5. Computational Studies
### 3.5.1. Hirshfeld Study

Hirshfeld surface (HS) analysis was performed to investigate the nature of the intermolecular interactions and their quantitative contributions to the crystal packing of compounds **1** and **2**. Figure 9 shows the HSs mapped over the $d_{\text{norm}}$ function for both compounds. The red spots on the surfaces (highlighted as dashed arrows) represent distances shorter than the sum of the vdW radii of atoms and the blue region corresponds to distances longer than the sum of vdW radii. Figure 10 shows the full two-dimensional fingerprint (FP) plots of **1** and **2**.

Compound **1**: The large red spots labeled 3 in the $d_{\text{norm}}$ map (Figure 9) are attributed to intermolecular O4W-H4WB···O7W involving the H4WB of the coordinated water molecule and the O7W of the non-coordinated water molecule as the acceptor. The large red regions labeled 1 and 2 in Figure 10 are mainly associated to O1W-H1WB···O3A and O4W-H4WA···O1A hydrogen bonds, respectively. The O12A and O11A of the uncoordinated *IPhth* anion are involved in two hydrogen bonding interactions with two coordinated water molecules, O5W-H5WA···O12A and O2W-H2WA···O11A. These H-bonds are visible on the HS mapped over the $d_{\text{norm}}$ function as bright red regions labeled 4 and 5. The presence of O3W-H3WA···O1A hydrogen bonds between one coordinated water molecule and the O1A of the uncoordinated *IPhth* anion is evidenced by the visible large red area labeled 6 in the $d_{\text{norm}}$ surface. The red spots labeled 7 on the HS are attributed to weak

C9-H9C⋯O11A involving the H9C of one methyl group of the coordinated *DMAP* ligand and the O11A of the uncoordinated *IPhth* anion as the acceptor. The proportion of H⋯O/O⋯H interactions comprise 37.0% of the total Hirshfeld surface area, and these contacts are identified as a pair of symmetrical spikes at $(d_e + d_i) \approx 1.70$ Å in the FP plots (Figure 10), which is consistent with the intermolecular distances reported in Table 3.

The presence of H⋯C/C⋯H contacts (6.30%) is identified by the presence of pronounced wings on both sides of the FP plots (Figure 10).

As was mentioned in the description of the crystal structure, the crystal packing of **1** is also stabilized by π⋯π stacking interactions involving the centroids of the C4A-C9A and the N1/C2-C6 rings, with an inter-centroid distance of 3.859 Å. In order to identify the π⋯π stacking interactions, the HSs were mapped with shape index and curvedness properties (Figure 11). The presence of red and blue triangles adjacent to each other (highlighted with a dashed circle) on the HS mapped over the shape index is indicative of π⋯π stacking interactions (Figure 11a). These contacts are also visible as relatively large and green flat regions indicated as a dashed arrow on the curvedness surface, as shown in Figure 11b. Finally, the C⋯C contacts involved in the π⋯π stacking interactions appear as a distinct pale blue to green area (highlighted as a red circle) in the FP plot (Figure 10) at $(d_e + d_i) \approx 3.80$ Å, with a contribution of 3.2% to the total Hirshfeld surface area.

Compound **1**

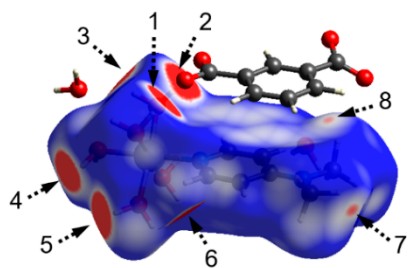

Compound **2**

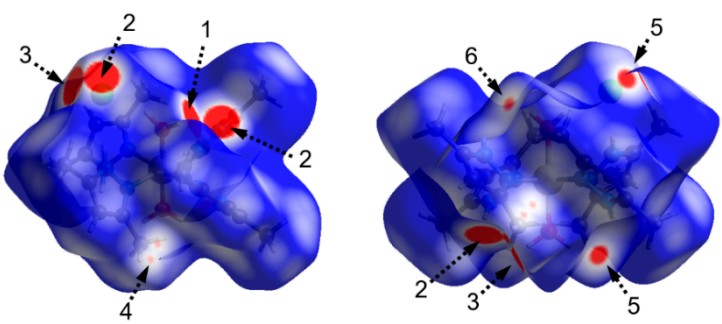

**Figure 9.** View of the Hirshfeld surfaces of **1** (top) and **2** (bottom) mapped over $d_{norm}$ property. For **2**, the second molecule is rotated by 180° around the horizontal axis of the plot. The labels are discussed in the text.

Compound **2**: The red regions labeled 1 and 3 in the $d_{norm}$ surface (Figure 9) are attributed to O2W-H2W⋯Cl1 and O1W-H1W⋯Cl1 involving the H-atoms of the coordinated water molecules and the chlorine anion as the acceptor. The two large red spots labeled 2 on the HS are mainly associated to N7-H7⋯Cl1 hydrogen bonds involving the N-H moiety of the pyrazole ring as the donor. The red regions labeled 5 and 6 in the $d_{norm}$ map are attributed to C10-H10A⋯Cl1 and C51-H51A⋯Cl1 hydrogen bonds, involving the H-atoms of the methyl group and the Cl1 anion as the acceptor. These H⋯Cl/Cl⋯H contacts are visible as sharp spikes (labeled 4) in the FP plots (Figure 10) at around $(d_e + d_i) \approx 2.00$ Å,

with a contribution of 14.3% to the Hirshfeld surface area. These contacts are visible in the $d_{norm}$ surface as small red spots labeled 4 and comprise 76% of the total HS area.

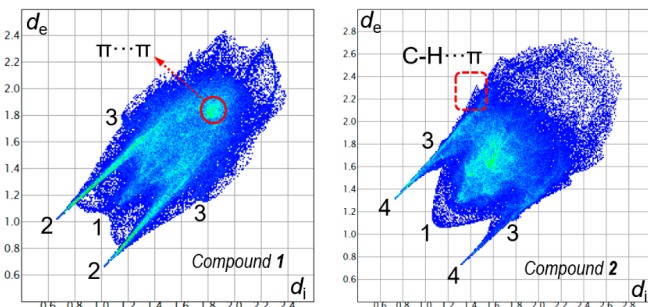

**Figure 10.** Full two-dimensional fingerprint plots for compounds **1** (left) and **2** (right) showing the spikes corresponding to the main intermolecular interactions. The percentage relative contributions of the intermolecular contacts to the total HS area are discussed in the main text. The FP plots show (1) H···O/O···H, (2) H···C/C···H and (3) H···Cl/Cl···H contacts.

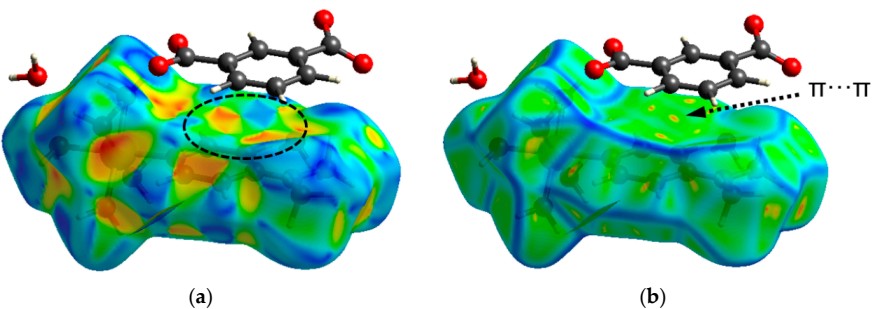

(**a**)           (**b**)

**Figure 11.** Hirshfeld surfaces of **1** mapped over shape index (**a**) and curvedness (**b**) properties.

The crystal packing of **2** shows C-H···π interactions involving the H9 atom of the pyrazole moiety and the N1/N2/C3-C5centroid [d(H4···Cg) = 3.417 Å]. These interactions are visible as a pair of wings in the top left and bottom right region in the FP plot (see Figure 10), which comprise 9.10% of the total Hirshfeld surface area. A view of the shape index confirms the existence of C-H···π interactions showing a large red depression above the pyrazole ring and the blue region surrounding the C9-H9 donor, with both regions highlighted as dashed arrows.

3.5.2. Theoretical Study

The theoretical study is focused on the π-stacked assemblies observed in compound **1** as a combination of anion-π and C-H···π interactions. Moreover, in these assemblies, the *IPhth* anions are H-bonded to the Ni(II) coordinated water molecules that werealso evaluated using the QTAIM analysis.

First, we computed the molecular electrostatic potential (MEP) surface to investigate the most electron-rich and -poor regions of compound **1**. We used the [Ni(H$_2$O)$_5$(DMAP)](IPhth) model (Figure 12) in order to use a charge-neutral model for the MEP calculations. It can be observed that the MEP values at the coordinated water molecules are very large (+84.7 kcal/mol) and correspond to the MEP maxima. The MEP is also positive over the ring center (+10.7 kcal/mol) and at the H-atoms of the dimethylamino groups (+13.8 kcal/mol). The MEP values are large and negative at the carboxylate groups (−62.7 and −55.2 kcal/mol) of *IPhth* and over the center of the aromatic ring (−37.7 kcal/mol, value not shown in Figure 12).

Figure 13 shows the combined QTAIM/NCI plot of the π-stacked trimer and the binding energy computed from the two possible dimers. The presence of two large RDG isosurfaces that embrace the whole space between the aromatic rings can be observed, thus

revealing a strong complementarity between the coordinated *DMAP* and the *IPhth* moieties. Moreover, a multitude of bond-critical points (CP, represented as small red spheres) and bond paths (represented as orange lines) further connect the aromatic rings and the substituents. In fact, the π-stacking assembly can be sub-divided into three interactions (anion–π, π–π and C–H⋯π), as indicated in Figure 13.

Moreover, an additional C–H⋯O H-bond is also observed. This intricate combination of interactions explains the large binding energies (−84.1 and −88.7 kcal/mol). Moreover, the combined QTAIM/NCI plot analysis discloses the existence of four OH⋯O H-bonds, which are characterized by a bond CP, bond path and small blue RDG isosurfaces interconnecting the H and O-atoms. The large interaction energies shown in Figure 10 are partially due to the ion-pair nature of the assembly. We also estimated the contribution of the H-bonds free from the pure Coulombic attraction by using the value of the Lagrangian Kinetic energy density (Gr) measured at the bond CP and the equation proposed by Veneret al. ($E_{ass} = 0.429 \times G_r$) [71] that was specifically developed for H-bonds in X-ray structures. The values are given in (Figure 13) close to the bond CPs and using a red font. The H-bonds are strong, with association energies ranging from −5.3kcal/mol to −6.6 kcal/mol, in line with the MEP surface analysis and the dark blue color of the RDG isosurfaces. The total contribution of the H-bonding interactions is −25.1 kcal/mol.

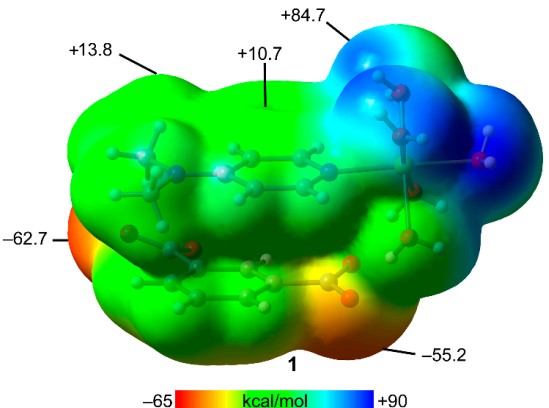

**Figure 12.** MEP surface of [Ni(H$_2$O)$_5$(DMAP)](IPhth) at the RI-BP86-D3/def2-TZVP level of theory (isosurface 0.001 a.u.). Energies in kcal/mol.

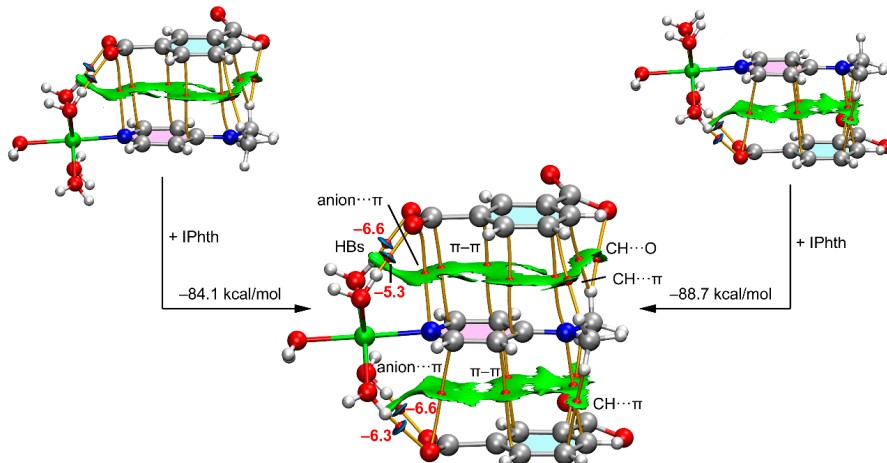

**Figure 13.** Combined QTAIM analysis (bond CPs in red, bond paths in orange) and NCI surfaces of two dimers of compound **1** (top) and the trimer (bottom). The gradient cut-off is ρ = 0.04 a.u., isosurfaces = 0.5, and the color scale is −0.04 a.u. < (signλ$_2$)ρ < 0.04 a.u. Only intermolecular interactions are shown. The energies of the H-bonds are given in red adjacent to the bond CPs in kcal/mol. The formation energy of the trimer from the two dimers is also indicated.

## 4. Conclusions

Two new pyridine- and pyrazole-based coordination compounds of Ni(II) and Co(II) were prepared and characterized using single-crystal X-ray diffraction, electronic, FT-IR spectroscopic techniques, elemental analysis and TGA. Several non-covalent contacts including anion–π, π–π, C–H···π, C–H···C, H···H along with H-bonding interactions stabilize the crystal structures. Self-assembled enclathration of counter chloride ions in compound **2** provides additional reinforcement to the crystal structure. All the non-covalent contacts were identified using Hirshfeld surface analysis. The studies reveal that both compounds **1** and **2** were stabilized by unconventional dihydrogen bonding contacts. Theoretical calculations on the supramolecular trimer observed in the crystal structure of **1** reveal that H-bonding interactions combined with the π-stacking contacts play an important role in the solid-state stability of compound **1**. The presence of two large RDG isosurfaces in the NCI plot analysis that embraces the whole space between the aromatic rings reveals the strong complementarity between the coordinated *DMAP* and lattice *IPhth* moieties.

**Supplementary Materials:** The following supporting information can be downloaded at: https://www.mdpi.com/article/10.3390/cryst13020203/s1, IR and UV-vis spectra of complexes **1** and **2** and additional description of the packing of **1** and **2**, thermogravimetric curves and additional references.

**Author Contributions:** Conceptualization, M.K.B. and A.F.; methodology, A.F. and D.M.G.; software, A.F. and D.M.G.; formal analysis, A.F. and D.M.G.; investigation, T.B. and R.M.G.; data curation, M.B.-O.; writing—original draft preparation, T.B., M.K.B. and A.F.; writing—review and editing, M.K.B. and A.F.; supervision, M.K.B. and A.F. All authors have read and agreed to the published version of the manuscript.

**Funding:** Financial supports from ASTEC, DST, Govt. of Assam (grant number: ASTEC/S&T/192(177) /2020–2021/43) and the Gobierno de España, Ministerio de Ciencia e Innovacion (project No. PID2020-115637GB-I00 FEDER funds) are gratefully acknowledged. The authors thank IIT-Guwahati for the TG data.

**Data Availability Statement:** Not applicable.

**Conflicts of Interest:** The authors declare no conflict of interest.

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
