# Peer review of "Supramolecular Assemblies in Pyridine- and Pyrazole-Based Coordination Compounds of Co(II) and Ni(II): Characterization, Hirshfeld Analysis and Theoretical Studies"

_crystals, doi:10.3390/cryst13020203_

Round 1

Reviewer 1 Report

In the presented manuscript Authors focused on the synthesis, crystal structures and properties of two new 3d-metal ion complexes. The current form of the manuscript cannot be accepted.

1)     Abstract and the text – Please explain what the term “lattice chloride ions” means. In the formulae of complex 2 the chloride ions there are outside of the coordination sphere so they are the counter anions.

2)     Introduction has to be rewritten because it is not present appropriate background for presented studies. Moreover it contains many repetitions.

3) Figure 4 does not scientific valuable and  should be removed from the manuscript.

4) Figure 6 should be corrected because molecular structure does not represent the formula of the compound 2 (one chloride ion is missed).

5) The discussion presented in the 3.4. Thermogravimetric analysis does not agree with the curves presented in Figure S6 (e.g. in the case of 1 the mass lost at the range 50-220 °C is below 20 % as well as the decomposition process undergoes in several steps). This part of the manuscript should be rewritten. Moreover, please explain the sentence “…Uncoordinated Cl moieties undergo decomposition …”. 

6) In the conclusion instead of Zn(II) should be Co(II).

7) Cif and checkcif files should be submitted for revision purpose.

Author Response

First, we would like to thank this reviewer for his/her careful reading of the manuscript, corrections and suggestions. Our point-by-point responses follow:

In the presented manuscript Authors focused on the synthesis, crystal structures and properties of two new 3d-metal ion complexes. The current form of the manuscript cannot be accepted.

Q1    Abstract and the text – Please explain what the term “lattice chloride ions” means. In the formulae of complex 2 the chloride ions there are outside of the coordination sphere so they are the counter anions.

Reply: We have replaced ‘lattice chloride ions’ with ‘counter chloride ions’ in the revised manuscript.

Q2     Introduction has to be rewritten because it is not present appropriate background for presented studies. Moreover it contains many repetitions.

Reply: We have revised the introduction section as suggested by the esteemed reviewer.

Q3 Figure 4 does not scientific valuable and should be removed from the manuscript.

Reply: We have removed Figure 4 from the revised manuscript.

Q4 Figure 6 should be corrected because molecular structure does not represent the formula of the compound 2 (one chloride ion is missed).

Reply: For compound 2, only one chloride ion is visible in the checkcif report, although there are two chloride ions present in the compound. The moiety formula of the compound (see Checkcif report) clearly indicates the presence of two chloride ions in the compound. Compound 2 possesses a two-fold axis of symmetry which passes through the coordinated water molecules (O1W, O2W) and the Co(II) centre of the compound. Recently, our research group has reported one Cu(II) compound viz. [Cu(bpy)2(NO3)]NO3·5H2O (where, bpy = 2,2´-bipyridine) [New J. Chem., 2021, 45, 8269-8282; DOI: 10.1039/d1nj01004b]. The asymmetric unit of the compound contains five lattice water molecules; however only three of them are visible in checkcif report as the two-fold axis of symmetry passes through the Cu1 centre and one of the lattice water molecules. Similarly for the compound [Cu22-C2O4)(phen)2(H2O)2][Cu(phen)(male)(NO3)]2 (phen = 1,10-phenanthroline, C2O4 = oxalate, male = maleate), only one [Cu(phen)(male)(NO3)] is present in the checkcif report [Inorg. Chim. Acta, 2019, 487, 354-361; https://doi.org/10.1016/j.ica.2018.12.037]. In this compound, the Cu(II) centers of the [Cu22-C2O4)(phen)2(H2O)2] unit lie on the two-fold axis of symmetry. Other research groups have also reported similar compounds where the metal centers lie on two-fold axis of symmetry [https://doi.org/10.1107/S2056989020003023; 10.1107/S1600536810028977; 10.1524/ncrs.2007.0032; 10.1021/ic00140a028].

Q5 The discussion presented in the 3.4. Thermogravimetric analysis does not agree with the curves presented in Figure S6 (e.g. in the case of 1 the mass lost at the range 50-220 °C is below 20 % as well as the decomposition process undergoes in several steps). This part of the manuscript should be rewritten. Moreover, please explain the sentence “…Uncoordinated Cl moieties undergo decomposition …”. 

Reply: We have revised the TG analysis accordingly.

Several other research groups have reported similar thermal behaviour of uncoordinated Cl (counter chlorine) in coordination compounds [J. Mol. Struct. 2009, 920, 149-162; 10.1016/j.molstruc.2008.10.059; Arab. J. Chem., 2014; 10.1016/j.arabjc.2014.10.013]. Our research group has also reported a few coordination compounds with similar thermal decomposition of uncoordinated Cl ion [J. Mol. Struct. 2021, 1229, 129486, https://doi.org/10.1016/j.molstruc.2020.129486; Inorg. Chim. Acta2021, 516, 120082-120093, 10.1016/j.ica.2020.120082]. We have now cited similar thermal decomposition of Cl in coordination compound in the revised mansucript.

Q6  In the conclusion instead of Zn(II) should be Co(II).

Reply: We have now corrected the mistake in the revised manuscript.

Q7 Cif and checkcif files should be submitted for revision purpose.

Reply: These files are provided. 

Reviewer 2 Report

The presented article describes the supramolecular structure of crystal samples of two nickel and cobalt coordination compounds. Various types of interactions are considered in detail, based on experimental structural data and various theoretical approaches. The work may be useful to the readers of the Crystals. It shows a variety of intermolecular interactions that determine the construction of crystal structures and which can be detected even in the most trivial objects. The paper can be published after some issues that have to be fixed:

1) The introduction is too extensive and covers a huge number of aspects of coordination and supramolecular chemistry. Authors should have focused more on the problems that are the direct objects of this study. The choice of specific objects that are not united by either the nature of the metal or organic ligands remains unclear. The metal complexes under discussion, judging by the initial reagents, probably became the products of unsuccessful attempts to synthesize the MOFs. In this vein, it would be interesting to trace the cause of such a failure just based on the beneficial formation of the detected structures.

2) The discussion of dihydrogen bonds should be completely excluded from the paper. There are two reasons for this. First, the quality of the X-ray diffraction experiment does not allow to reliably localize protons in structures. And second, hydrogen bonds are formed between protons and hydrides, which are absent in the studied structures.

3) Lines 385-387. Uncoordinated chloride can decompose only during nuclear transformation. This phrase needs to be corrected.

Author Response

We thank Reviewer 2 for his/her careful reading of the manuscript, corrections and suggestions. Our “point-by-point” responses follow:

The presented article describes the supramolecular structure of crystal samples of two nickel and cobalt coordination compounds. Various types of interactions are considered in detail, based on experimental structural data and various theoretical approaches. The work may be useful to the readers of the Crystals. It shows a variety of intermolecular interactions that determine the construction of crystal structures and which can be detected even in the most trivial objects. The paper can be published after some issues that have to be fixed:

  • The introduction is too extensive and covers a huge number of aspects of coordination and supramolecular chemistry. Authors should have focused more on the problems that are the direct objects of this study. The choice of specific objects that are not united by either the nature of the metal or organic ligands remains unclear. The metal complexes under discussion, judging by the initial reagents, probably became the products of unsuccessful attempts to synthesize the MOFs. In this vein, it would be interesting to trace the cause of such a failure just based on the beneficial formation of the detected structures.

Reply: We have now removed the irrelevant discussions and revised the introduction part of the manuscript focusing on the relevant organic ligands to justify the synthesis of the reported two new coordination compounds. However, this can be considered as a successful attempt to design metal-organic compounds (coordination compound) with the pyridine and pyrazole based ligands.

  • The discussion of dihydrogen bonds should be completely excluded from the paper. There are two reasons for this. First, the quality of the X-ray diffraction experiment does not allow to reliably localize protons in structures. And second, hydrogen bonds are formed between protons and hydrides, which are absent in the studied structures.

Reply: The discussion of dihydrogen bonds have been completely excluded from the manuscript.

3) Lines 385-387. Uncoordinated chloride can decompose only during nuclear transformation. This phrase needs to be corrected.

Reply: We have now removed the word ‘uncoordinated’ from the revised manuscript. Several other research groups have reported similar thermal decomposition of uncoordinated Cl in coordination compounds [J. Mol. Struct. 2009, 920, 149-162; 10.1016/j.molstruc.2008.10.059; Arab. J. Chem., 2014; 10.1016/j.arabjc.2014.10.013]. Our research group has also reported a few coordination compounds with similar thermal decomposition of uncoordinated Cl ion [J. Mol. Struct. 2021, 1229, 129486, https://doi.org/10.1016/j.molstruc.2020.129486; Inorg. Chim. Acta 2021, 516, 120082-120093, 10.1016/j.ica.2020.120082].

Author Response

We thank Reviewer 3 for his/her careful reading of the manuscript, corrections and suggestions. Our “point-by-point” responses follow:

Generally, the idea of the research is interesting. Two structures of coordination compounds of Ni(II) and Co(II) involving pyridine and pyrazole moieties have great potential in supramolecular analysis.

Nevertheless, there are some points in the presented work that need to be explained or considered more carefully.

Q1. First of all, the structural data are poor quality. The completeness of the X-ray data is only 90 %. The yield of the synthesis of (1) was declared to be 88%. The monocrystal was quite big with dimensions 0.48x0.28x0.25 [mm]. Did the authors try to find another crystal ? In case of cif files, authors should consider many alerts in checkcif reports to present the data more accurately. Treatment of H-atoms in water molecules should be examined more carefully. Some examples: the strange parameter of Uiso=10.08 (blocked) declared for the independently refined H atoms H7WA and H7WB in structure (1). The remaining Uiso for H atoms in water molecules are declared to be -1.5Ueq(O). Moreover, why _refine_ls_shift/su_max is equal 0.0 if the value is always in .cif given as 0.000 (it looks like the value is cut by the authors). The last mentioned parameter determines the convergence of the calculations.

We have checked the data reduction of the crystal and found a mistake in the cell. After a new data reduction and solution, now the crystal resolution is much better. We think that now there is not any major issue. A new cif has been generated

Q2. During the refinement of (2), six reflections have been rejected from the refinement; please justify it.

Reply: We have re-refined the structure. The bond precision is low, but there are no A or B alerts, only C-alerts. After a careful revision, it seems that it is not a major issue. Regarding the omitted reflections, now just two of them are omitted, which are probably affected by the beam stopper. The two gave an error/esd ratio higher than 10 and appeared completely out of trend in the Fobs/Fcalc graph

Q3. The bond precision of of 0.0083Å is low in structure (2), in structure (1) is better ‘maybe by chance’ (0.0046Å) since the completeness of the data is poor and the numerous water molecules in structure (1) are determined with constraints / restrictions. In this context, in my opinion, many hydrogen bonds considered in the manuscript are not reliable, especially H…H interactions. In the context of the remarks given above, the re-refinement of the X-ray data should be done to fulfil the IUCr requirements (please see checkcif alerts, i.e. how the short H…H contacts are reliable under normalised procedure). Usually, short H…H contacts show potential problems in determining the crystal structures.

Reply: We have now removed the H···H interactions in the revised manuscript as also pointed out by esteemed reviewer 2. 

The comments, from the beginning, step by step in the manuscript:

  1. The abbreviation DMAP has been expanded once in the whole manuscript, in the abstract and is wrong. DMAP – is 4-dimethylaminopyridine instead of 4-dimethylpyridine. Once the proper name is given on page 3 (without the abbreviation).

Reply: We have corrected the mistake in the abstract of the revised manuscript.

  1. Table 2 – the temperature of the X-ray measurement for (1) is given as 151K in the cif file (the temperature of refined data is important for the proper determination of the lengths of H-X bonds for H-atoms in rigid body model) but declared as 294K in Table 2.

Reply: The corrections have been done in the revised manuscript as pointed out by the esteemed reviewer.

  1. Table 2 – the number of observed reflections with a criterion of observation should be given.

Reply: This has been included in the revised manuscript.

  1. Figure 1 - Labels for atoms in the pyridine moiety should be given.

Reply: Atom labeling has been done for the Figure as suggested.

  1. Table 2 – some values are wrong i.e. the angle N1-Ni1-O3W=2.46(1°) instead of 92.46(1)° Please correct the values for valence angles: N1-Ni1-O5W, N1-O1W, N1-Ni1-O4W.

Reply: The corrections have been done in the revised manuscript.

  1. Line 225 (page 7) C8…Cg distance should be given with s.u. Moreover, the C8-H8A..Cg should be given. From the calculations it is equal to 108 angle, and this angle does not support the existence of CH…π interaction

Reply: Distances have been incorporated in the revised manuscript as suggested by the esteemed reviewer. We have also incorporated the corresponding C8-H8A···Cg angle. The existence of C-H···π interaction has been confirmed by the combined QTAIM/NCI plot computational tools.

  1. Line 230 (page 7) – the ‘strong’ anion-p interaction should be justified. The X-Cg distance. (where X=O3A) should be given with s.u. How the angle between O3A and the centroid of aromatic plane (Cg) was found as equal to 95.64.° if the angle should be defined by three atoms° (from Platon this angle for C2-O3A …Cg = 66.66(14) interaction should be described more precisely.

Reply: The distance has been given with s.u. for the anion-p interaction. In anion-p interaction, the angle between the anion, ring centroid and the aromatic plane of the ring is considered (CrystEngComm. 2011, 13, 4519-4527; https://doi.org/10.1039/C0CE00593B). In this case, the angle between O3A, Cg (ring centroid) and C2 atom (defining the aromatic plane) is found to be 95.65°, which can be observed in Diamond software. We have now revised the text more precisely for the interaction in the revised manuscript.

  1. Lines 134-239 (page 7) the p-p interaction should be described more precisely. All parameters involving non-hydrogen atoms should be given with s.u.

Reply: Parameters have been given with s.u. for the p-p interaction and the corresponding slipped angle.

  1. Figure 4 and the corresponding description to Table 3. The table of potential H-bonds in cif files does not correspond to the results in Table 3 of the manuscript. Please explain the difference? It is very important in the description of the crystal packings of (1) and (2). To show that the data are misleading: the angle of interaction O7W-H7WB…O6W in cif file and checkccif , but in the Table 3 the data is completely inconsistent as given equal to158.9°, and O-H bond length is 0.85Å (cif) in contrast to 0.87Å (in the manuscript), the corresponding H..A distance differs from reported 1.98Å to 2.69Å from cif. To be honest, there is no interaction presented in Figure 4 and therefore the further descriptions (conclusions) are wrong.

Reply: For Figure 4; O7W‒H7WB∙∙∙O6W hydrogen bonding is associated with O7W‒H7WB distance of 0.87 Å, O7W∙∙∙O6W distance of 2.813 Å, H7WB∙∙∙O6W distance of 1.98 Å and H7WB∙∙∙O6W angle of 158.9°. All these values are as per the packing diagram made in Diamond software. Even the mercury software also represents these interactions with the aforementioned parameters.

  1. Table 3 – the intermolecular C-H…O interactions with angles smaller than 120 have no special meaning, i.e. 113°, 107 interactions, especially with long H..A distance. Please reconsider such types of interactions

Reply: We have removed these two C-H···O interactions in the revised manuscript.

  1. Table 3 - All interactions are given without symmetry codes between the interacting groups (atoms) . In crystallographic descriptions, the symmetry is the most important.

Reply: We have incorporated the symmetry codes.

  1. Figure 6 – no labels for ligand atoms.

Reply: Labeling has been done as suggested.

  1. Most of the remarks given above for structure (1) are also working for structure (2).

Reply: We have done all the modifications for compound 2 as suggested.

  1. The theoretical calculations are based on the geometries from crystal structures. First of all, the geometries are not determined accurately for H-atoms in water molecules (geometrical constraints). Secondly, for theoretical single-point calculations the normalisation of Hpositions must be done; there is no any short paragraph on such a procedure used for analysed structures; and the X-H bond lengths used for the theoretical model are not shown.

Reply: The position of the H-atoms was optimized in the DFT calculations. This has been now clarified in the theoretical methods

  1. The evaluation of energy interaction based on the equation given by EML [Espinosa, E.; Molins, E.; Lecomte, C. Hydrogen bond strengths revealed by topological analyses of experimentally observed 724 electron densities. Chem. Phys. Lett. 1998, 285, 170-173] was judged and found to be misleading by M.A. Spackman [Cryst. Growth. Des. 2015, 15, 5624-5628], and in this context the whole energetic analysis at the bond critical point is useless without additional support of energetic calculations, for example performed with PIXEL.

Reply: Thank you, we have now used the procedure reported by Vener et al., developed for X-ray structures, see: Vener, M.V.; Egorova, A.N.; Churakov, A.V.; Tsirelson, V. G. J. Comput. Chem. 2012, 33, 2303-2309

Round 2

Reviewer 1 Report

The manuscript has been sufficiently improved to warrant publication in Crystals. The aatached

Author Response

The manuscript has been sufficiently improved to warrant publication in Crystals.

Reply: Thank you

Author Response

We thank the reviewer for his/her second reading of the manuscript, corrctions and suggestions. Our responses follow

The following remarks should still be considered:

  • In Table 2 for compound 2 the symmetry of primed atoms is required (given as footnotes); the same is true for Figure 5 (given in the caption of the figure).

Reply: We have incorporated the symmetry of primed atoms as suggested by esteemed reviewer.

  • In Table 3, most of the symmetry codes for the intermolecular hydrogen bonding for structures 1 and 2 are still missing. The authors added symmetry codes only for four ‘selected’ interactions for structure 1. Why?

Reply: We have now added the symmetry codes observed in the CIFs of the compounds. Also, to make Table 3 consistent with the CIFs of the compounds; the H-bonding interactions observed  in the crystallographic software (Diamond 3.2) have been omitted in the table.

  • In Table 3, the donor…acceptor distances should be given with e.s.d. because non-H atoms are refined in comparison to H-atoms.

Reply: We have incorporated the e.s.d. values in Table 3.

Important: In Table 3, for structure 2: hydrogen bonds should be examined carefully once again. According to the description in lines 121-123 that hydrogen atoms of water molecules in structure 2 are refined freely, parameters with those H atoms should be given with esd.

Reply: We have now added the e.s.d. values for structure 2 in Table 3 of the revised manuscript.

It is strange that the geometric parameters are completely different than those in cif; especially see C-H and O-H bonds etc. One can understand that pictures describing crystal packing can be done almost automatically by crystallographic software, but the data in tables should be consistent with cif, symmetry related to asymmetric unit, etc.

Reply: We have revised Table 3 accordingly. We thank the esteemed reviewer for the suggestion.

By the way, personally, I am not convinced to some C-H…O interactions described in the manuscript (as for example C8-H8C…O1A at line 200 or C9-H9C…O1A (table 3) for which the H…O contacts are longer than the sum of van der Waals radii of the corresponding atoms.

Reply: This C–H···O interaction has been further supported using combined NCI plot/ QTAIM computational tool (Figure 13). The study reveals the presence of a bond CP, bond path and extended green isosurfaces interconnecting the H and O-atoms; thereby supporting the presence of the hydrogen bonding interaction. Moreover, the Hirshfeld surface (HS) analysis (Figure 10) also supports the presence of this C–H···O hydrogen bonding interaction.